# Guar Gum as an Eco-Friendly Corrosion Inhibitor for N80 Carbon Steel under Sweet Environment in Saline Solution: Electrochemical, Surface, and Spectroscopic Studies

**DOI:** 10.3390/ijms241512269

**Published:** 2023-07-31

**Authors:** Gaetano Palumbo, Dominika Święch, Marcin Górny

**Affiliations:** 1Department of Chemistry and Corrosion of Metals, Faculty of Foundry Engineering, AGH University of Science and Technology, al. Mickiewicza 30, PL-30059 Krakow, Poland; dswiech@agh.edu.pl; 2Department of Cast Alloys and Composites Engineering, Faculty of Foundry Engineering, AGH University of Science and Technology, al. Mickiewicza 30, PL-30059 Krakow, Poland; mgorny@agh.edu.pl

**Keywords:** guar gum, CO_2_ corrosion, corrosion inhibition, carbon steel N80

## Abstract

In this study, the corrosion inhibition performance of the natural polysaccharide guar gum (GG) for N80 carbon steel in CO_2_-saturated saline solution at different temperatures and immersion times was investigated by weight loss and electrochemical measurements. The results have revealed that GG showed good inhibition performance at lower and higher temperatures. The inhibition efficiency observed via weight loss measurements reached 76.16 and 63.19% with 0.4 g L^−1^ of GG, at 25 and 50 °C, respectively. The inhibition efficiency of GG increased as the inhibitor concentration and immersion time increased but decreased with increasing temperature. EIS measurements have shown that, even after prolonged exposure, GG was still able to protect the metal surface. Potentiodynamic measurements showed the mixed-type nature of GG inhibitive action. The Temkin and Dubinin–Radushkevich adsorption isotherm models give accurate fitting of the estimated data, and the calculated parameters indicated that the adsorption of GG occurred mainly via an electrostatic or physical adsorption process. The associated activation energy (*E*_a_) and the heat of adsorption (*Q*_a_) supported the physical adsorption nature of GG. FTIR analysis was used to explain the adsorption interaction between the inhibitor and the N80 carbon steel surface. SEM-EDS and AFM confirmed the adsorption of GG and the formation of an adsorptive layer of GG on the metal surface.

## 1. Introduction

CO_2_ is one of the greenhouse gases accountable for causing global warming. The majority of CO_2_ emissions in the atmosphere come from the combustion of fossil fuels, industrial processes, agricultural practices, etc. [1,2]. CO_2_-enhanced oil recovery (CO_2_-EOR) is one of the most favorable ways to decrease the accumulation of CO_2_ emissions in the atmosphere. The CO_2_-EOR process consists of injecting CO_2_ into a reservoir to improve oil production [3,4]. Some portion of the CO_2_ injected remains in the reservoir (i.e., below the ground) while the portion that returns to the surface is reinjected into the reservoir, thus creating a closed loop that leads to permanent CO_2_ storage. As a result, the crude oil obtained from the CO_2_-EOR processes is commonly labeled as “carbon-negative oil” [2,5]. However, CO_2_ that has been dissolved in the fluid can be a serious problem for steel-based facilities [4,6,7,8]. Moreover, the corrosion attack becomes more severe in the presence of high chloride concentration in the fluid [4].

N80 carbon steel is very often used in the oil and gas industry due to its mechanical properties and low cost [4,6,8,9]. Nevertheless, carbon steel is vulnerable to the risk of sweet corrosion because it does not develop a protective film on its surface as stainless-steel alloys do. The use of high corrosion-resistant alloys and/or corrosion inhibitors are two important methods to mitigate sweet corrosion. The first method can be quite expensive if considering the large-scale reservoir facilities, whereas the second is generally regarded as one of the most cost-effective viable methods of reducing the threat of sweet corrosion. A corrosive inhibitor protects the metallic infrastructures by forming an adsorptive layer that acts as a barrier between the aggressive environment and the steel surface. According to a literature survey, nitrogen-based compounds such as amine derivatives are widely used in the petroleum industry as corrosion inhibitors [10]. However, these compounds are reported to be toxic and harmful to the environment [10]. Moreover, due to the increasingly stringent environmental regulations, the use of naturally occurring corrosion inhibitors has gained great attention in this regard. Umoren et al. [11] evaluated the effect of chitosan and carboxymethyl cellulose (CMC) as eco-friendly corrosion inhibitors for API 5L X60 steel in a CO_2_-saturated saline solution. The authors observed that these natural compounds were able to protect the steel from sweet corrosion. Palumbo et al. [4] studied the corrosion inhibition effect of gum arabic in a saturated CO_2_-saline environment. The study reported that this natural polysaccharide acted like a good corrosion inhibitor for N80 carbon steel with a maximum inhibition of circa 70% after 24 h of immersion. Obot et al. [12] evaluated the polysaccharide sodium alginate for preventing the corrosion of API X60 carbon steel in 3.5% NaCl and reported a maximum inhibition efficiency of 87.23%.

Guar gum (GG) is a water-soluble, biodegradable, and naturally occurring polysaccharide extract from the endosperm of the Cyanmopsis tetragonolobus belonging to the family Leguminosae. This compound is widely used in many industries, such as biomedical, wastewater treatment, and food [13]. Abdallah [14] was the first to test GG as a possible corrosion inhibitor for carbon steel in H_2_SO_4_ solutions. The author observed a maximum value of the corrosion efficiency of circa 93.88% at 25 °C. After this first study, many other studies have been published on the use of GG as a corrosion inhibitor for different materials and environments, and all of them agreed on the efficacy of GG as a corrosion inhibitor [7,13,14,15,16,17,18]. In the continuation of our research effort toward the study of green corrosion inhibitors to mitigate sweet corrosion, we report for the first time the inhibition effect of guar gum as an eco-friendly corrosion inhibitor for N80 carbon steel in a CO_2_-saturated saline solution at different temperatures for the oil and gas industry. All of the studies that have been carried out so far with GG as a corrosion inhibitor were performed in strong acid solution (e.g., 1 M H_2_SO_4_ [14], 2 M H_3_PO_3_ [13], 1 M HCl [18], etc.) and for short immersion times. The corrosion of the steel in the presence of CO_2_ can be more complex. The formation of iron carbonate on the metal surface can influence the corrosion resistance of the steel. Therefore, understanding this type of corrosion is important for corrosion engineering in the oil and gas field. Furthermore, GG is already broadly used in drilling fluids as a thickening agent [19]. The objective of this manuscript is to demonstrate that GG not only can be used as a thickening agent in drilling fluids, but it could also be used as an active component to mitigate sweet corrosion in the oil and gas reservoirs.

To this end, the anticorrosive properties of guar gum were studied via weight loss and electrochemical measurements with different concentrations of GG and different temperatures. The adsorption mode of GG on the metal surface (i.e., adsorption isotherm and activation parameters) was calculated and discussed. Fourier-transform infrared spectroscopy (FTIR), scanning electron microscopy (SEM–EDS), and atomic force microscopy (AFM) analysis were also employed to characterize the metal surface and to support the gravimetric and electrochemical findings.

## 2. Results and Discussion

### 2.1. Effect of Concentration and Temperature

#### 2.1.1. Weight Loss Measurements

The calculated corrosion rate (CR) and inhibition efficiency (IE%) values using the weight loss methods at varying concentrations of GG and temperatures after immersion for 24 h in the tested solution are reported in Figure 1a,b, respectively. Notably, the CR of N80 carbon steel depends on the concentration of GG in both temperatures. The values of IE% increase with the GG amount, which corresponds to the adsorption of GG on the metal surface. However, it is interesting to note that the performance of GG varies greatly with temperature. Two distinct behaviors can be observed at 25 and 50 °C. For instance, as the temperature increases, the IE% values decrease, which is indicative that GG is physically adsorbed on the steel surface [13,15]. Furthermore, at lower temperatures, the values of IE% have already attained a stable value after the addition of 0.2 g L^−1^ of GG (e.g., 74.55%), whereas at higher temperatures, the value of IE%, continues to slowly increase with the GG concentration. The loss in the inhibitive performance of GG with increasing temperatures might be owing to its desorption from the metal surface, even though it is worth noting that GG is still able to mitigate sweet corrosion to some extent, even at a higher temperature (i.e., 63.16%). The good IE% value observed at higher temperatures and its optimum value that was already reached at low concentrations of GG can be directly linked to its solubility, which increases with temperature, and with the physicochemical properties of GG [20,21]. When guar gum is dissolved in water, it forms a highly viscous solution even at a low concentration. This effect can be ascribed to the strong hydrogen bonds formed between the macromolecules, which leads to a high degree of inter-molecular chain entanglement in the inhibitor macromolecules. As the concentration of GG increases, the amount of inter-molecular chain interaction increases, which increases viscosity [22]. In this context, the inter-molecular chain entanglement reduces their mobility and prevents them from moving from the bulk of the solution to the steel surface, thereby reducing the possibility that they will be absorbed on the surface. However, at higher temperatures, due to the weakening of these inter-molecular bonds (e.g., hydrogen bonds are weaker with an increase in temperature) they attain a certain degree of mobility, which results in an increase in IE% at higher temperatures [20]. Another phenomenon that must be taken into account is that GG in solutions is stable for short-term heating, but degrades when it is warmed for longer times [23]. Its degradation leads to the formation of shorter and less entangled chain segments with higher mobility, compared to the full macromolecules, which increases their probability of being absorbed on the metal surface.

Table 1 lists the inhibition efficiency and the time of exposure at which the maximum inhibition efficiency has been calculated for various natural polysaccharides corrosion inhibitors. It follows from the table that these studies were mostly performed at short immersion times (i.e., up to 6 h), and only a few of them were carried out at long immersion (i.e., 24 h). The table shows that, compared to other natural corrosion compounds, GG can be considered a good environmentally friendly corrosion inhibitor for carbon steel in a CO_2_-saturated chloride solution even after prolonged exposure times.

#### 2.1.2. Adsorption Study and Standard Adsorption Free Energy

The adsorption behavior of guar gum on the N80 carbon steel surface was studied by various adsorption isotherm models. The best fit was obtained with Langmuir (Equation (1)) [15,26], Temkin (Equation (2)) [4,15], and Dubinin–Radushkevich (Equation (3)) [26,27], as shown in Figure 2.
(1)$$\frac{C_{\mathrm{inh}}}{\theta }=\frac{1}{K_{\mathrm{ads}}}+C_{\mathrm{inh}}$$
(2)$$\theta =-\frac{\ln K_{\mathrm{ads}}}{2a}\frac{\ln C_{\mathrm{inh}}}{2a}$$
where *K*_ads_ is the constant related to adsorption energy, *C*_inh_ is the concentration of the inhibitor, and *a* is the adsorbate interaction factor [26,27].
(3)$$ln\theta =\ln\theta_{\max}-\varepsilon \delta^{2}$$
where *θ*_max_ is the maximum surface coverage and *ε* is Polanyi potential. The Polanyi potential is related to the inhibitor concentration by the following Equation (4) [26,27]:(4)$$\delta =RT\;\ln\left(1+\frac{1}{C_{\mathrm{inh}}}\right)$$
where *R* is the universal gas constant and *T* is the absolute temperature. The constant *ε* is used to estimate the mean adsorption energy (*E*) which represents the energy used in transferring 1 mol of the inhibitor from the bulk solution toward the metal surface. The value of *E* was calculated with the following Equation (5) [26,27]:(5)$$E=\frac{1}{\sqrt{2\varepsilon }}$$
*E* provides information about the physicochemical characteristics of the adsorption process. If the value of *E* is <8 kJ mol^−1^, this indicates physical adsorption, and if its value is >8 kJ mol^−1^, this indicates that the adsorption occurs via a chemical process [23,24]. *K*_ads_ calculated from the adsorption isotherms can be linked to the free energy of adsorption (${∆G}_{\mathrm{ads}}^{{}^{\circ}}$) by the following relationship (Equation (6)) [4,7,13,15]:(6)$${∆G}_{\mathrm{ads}}^{{}^{\circ}}=-RT\;\ln\;(1\;\times {\;10}^{3}\;K_{\mathrm{ads}})$$
where the value 1 × 10^3^ is the concentration of water molecules in solution expressed in g L^−1^. 

The data listed in Table 2 shows negative values of ${∆G}_{\mathrm{ads}}^{{}^{\circ}}$, which indicates that the adsorption of GG is a spontaneous process [4,13,15,28]. The values of ${∆G}_{\mathrm{ads}}^{{}^{\circ}}$ and *E* indicate that the adsorption process of GG may be mainly a mixed interaction (i.e., physical and chemical adsorption), although the combination of electrostatic interaction and physical sorption is more dominant. Moreover, the values of *a* are found to be negative in both tested temperatures, which is indicative of repulsive forces between the adsorbed molecules [4,13,15]. The findings are in agreement with previous studies [4,13,15,28,29].

The dependence of the corrosion rate on temperature can be expressed by the condensed Arrhenius equation (Equation (7)) [4,9,15,29,30]:(7)$$\log\frac{{\mathrm{CR}}_{2}}{{\mathrm{CR}}_{1}}=\frac{E_{a}}{2.303\;R}\left(\frac{1}{T_{1}}-\frac{1}{T_{2}}\right)$$
where CR_1_ and CR_2_ are the values of the corrosion rate at temperature *T*_1_ (25 °C) and *T*_2_ (50 °C), respectively. *E*_a_ is the apparent activation energy. The values of heat of adsorption were calculated with the following Equation (8) [4,9,15,29]:(8)$$Q_{ads}=2.303R\left[log\left(\frac{\theta_{2}}{1-{\;\theta }_{2}}\right)-log\left(\frac{\theta_{1}}{1-{\;\theta }_{1}}\right)\right]\times \left(\frac{T_{1}\times T_{2}}{T_{2}-T_{1}}\right)$$
where *θ*_1_ and *θ*_2_ are values of the degree of surface coverage at the tested temperatures. The calculated parameters are listed in Table 3.

It is generally believed that, if the values of the activation energies are less than 40 kJ mol^−1^, the process can be regarded as a physical adsorption process, while values higher than 80 kJ mol^−1^ point toward a chemisorption adsorption process. Furthermore, for values of *Q*_ads_ between −20 and 0 kJ mol^−1^, the adsorption process is considered to be a physical process, but with values between −400 and −80 kJ mol^−1^, the process is regarded as a chemical process [9,13,15,29,31]. From the table, it is clear that the values of *E*_a_ and *Q*_ads_, combined with the trend of the decrease in inhibition efficiency with an increase in temperature, further support the physical adsorption nature of the process proposed.

#### 2.1.3. Electrochemical Measurements

Figure 3 shows the potentiodynamic polarization curves of N80 carbon steel after 24 h of immersion in CO_2_-saturated saline solution in the absence and presence of different concentrations of GG and different temperatures. The electrochemical parameters, such as the corrosion potential (*E*_corr_), the cathodic (*β*_c_) Tafel slopes, the corrosion current density (*i*_corr_), and inhibition efficiency (IE%), are shown in Table 4. Notably, the anodic branches do not show the typical Tafel region; as a result, the corrosion current densities were calculated from the extrapolation of the cathodic Tafel region according to Amin et al. [32].

In an aqueous solution, CO_2_ gas dissolves to form carbonic acid [33,34]: (9)$$CO_{2}+H_{2}O↔H_{2}CO_{3}$$
which in turn it dissociates into bicarbonate and then carbonate anions:(10)$$H_{2}CO_{3}↔H^{+}+\mathrm{HC}O_{3}^{-}$$
(11)$$\mathrm{HC}O_{3}^{-}↔H^{+}+CO_{3}^{2-}$$
which lowers the pH leading to the following cathodic reactions [33,34]:(12)$${2H}^{+}+2e^{-}\to H_{2}$$
(13)$$2H_{2}CO_{3}+2e^{-}\to H_{2}+2\mathrm{HC}O_{3}^{-}$$
(14)$$2\mathrm{HC}O_{3}^{-}+2e^{-}\to H_{2}+2CO_{3}^{2-}$$
(15)$$2H_{2}O+2e^{-}\to H_{2}+2OH^{-}$$

In a CO_2_-saturated solution, the anodic reaction of the steel is a multi-step process (a consecutive mechanism) [33,34]:(16)$$\mathrm{Fe}+H_{2}O_{}\to {\mathrm{FeOH}}_{\left(\mathrm{ads}\right)}+H^{+}+e^{-}$$
(17)$${\mathrm{FeOH}}_{\left(\mathrm{ads}\right)}\to \;\mathrm{FeO}H^{+}+e^{-}$$
(18)$$\mathrm{FeO}H^{+}+H^{+}\to {\mathrm{Fe}}^{2+}+H_{2}O_{}$$

It follows from the data that *i*_corr_ decreases by one order of magnitude after the addition of 0.1 g L^−1^ of GG for both temperatures. This behavior is likely associated with the formation of a protective film on the steel surface that hinders the charge–transfer reactions and, consequently, its dissolution. As observed in Figure 3, both cathodic and anodic reactions were slowed down by the addition of GG, and *E*_corr_ did not change with and without the presence of GG. These results indicate that GG acts as a mixed-type inhibitor in agreement with the results previously reported for GG [7,13,14,15]. By looking more closely at Figure 3 and the values of *β*_c_, it can be seen that the cathodic curves with and without GG are different. The cathodic curve observed in the blank solution shows a more diffusion control type of behavior whereas, after the addition of GG, it shows a more linear, Tafel-type behavior. This behavior has been reported in several studies, and it might be due to the non-homogeneous reaction of CO_2_ hydration on the metal surface in a CO_2_-saturated environment [9,30,33]. This observation indicates that GG influences the mechanism of the corrosion process by preventing and/or inhibiting the CO_2_ hydration reaction on the metal surface [9,30,33].

The dominant cathodic reaction depends on the pH of the solution [33,34]; at pH < 4, Equation (12) is the dominant one. At pH between 4 and 6, Equations (13) and (14) become the dominant cathodic reactions. Finally, at a high pH, the dominant cathodic reaction is the reduction of water, Equation (15). For the anodic reactions, the rate-determining step is Equation (17). In this study, the pH value of the solution was found to be 4.0, which indicates that GG was able to hinder both the anodic reaction and cathodic reduction of H^+^ and H_2_CO_3_ [34]. In an acid solution, GG can be protonated; therefore, it exists as a cationic species. Its cationic form competes with the H^+^ and H_2_CO_3_ for adsorption onto cathodic sites on the steel surface. After adsorption, its large size, compared to H^+^ and H_2_CO_3_, isolates these reactive sites, decreasing the rate of transfer of electrons from anodic to cathodic sites on the metal surface [7,13], as will be discussed in more detail in Section 2.4.

The data also shows that *i*_corr_ further decreases upon a further increase in GG concentration, which also corresponds to a maximum inhibition efficiency value of 86.51 and 81.04% after 24 h of immersion time at 25 °C (0.3 g L^−1^ of GG) and 50 °C (0.4 g L^−1^ of GG), respectively. The findings indicate that the adsorption of GG increases the surface coverage on the steel surface, thereby blocking more and more active sites in the anodic and cathodic regions.

Figure 4 shows the Nyquist and Bode plots carried out at 25 °C (Figure 4a,b), and 50 °C (Figure 4c,d), respectively. It can be seen from the figures that, with or without the presence of GG, all the impedance spectra are characterized by broad depressed semicircles at high (HF) to medium frequency (MD). An inductive loop at low frequency (LF) can also be seen in the blank solution and at lower concentrations of GG (e.g., up to 0.1 g L^−1^). The capacitive loop at HF has been associated with the electron transfer process on the surface as well as the charge–discharge process of the electrical double layer between the electrode surface and electrolyte solution [34]. While the inductive loop could be ascribed to the relaxation process caused by the adsorption of intermediate species on the metal surface, such as FeOH_(ads)_ (Equations (16)–(18)), $Cl_{\left(\mathrm{ads}\right)}^{-}$, and $H_{\left(\mathrm{ads}\right)}^{+}$, as reported by many studies [9,32,34,35,36]. When the concentration of GG increases, the depressed semicircle grows in diameter and the inductive loop gradually disappears. The disappearance is most likely due to its shift to lower frequencies that cannot be seen in the range of the studied frequencies [35]. However, as the concentration of GG increases, the disappearance of the inductive loop is followed by the appearance of a second-time constant in the HF region (i.e., between 10^4^ and 10^2^ Hz, from the phase angle diagram), along with the time constant of the charge-transfer process and the double-layer capacitance. Similar results were observed by several authors [3,8,28,35] and could be explained by the adsorption of the inhibitor and the formation of an adsorptive inhibitor layer that reduces the rate of the cathodic and anodic reactions.

The EIS plots were fitted with electric equivalent circuits (EEC) with two-time constants, one for the uninhibited and low concentrations of GG [8,28,35] (Figure 5a), and one for higher concentrations of GG [3,4,6,35] (Figure 5b). Where *R*_s_ is the solution resistance, *R*_ct_ and CPE_dl_ are the charge transfer resistance and the constant phase element representing the double-charge layer capacitance, respectively. *R*_L_ and *L* are the inductive resistance and inductance, respectively. *R*_f_ and CPE_f_ are the resistance provided by the adsorbed inhibitor layer and its constant phase element, respectively.

The depressed semicircles are an indication of an inhomogeneous metal surface, and in such cases, a “constant phase element” (CPE) is used instead of the double-layer capacitance (*C*_dl_). The impedance of a CPE is defined by Refs. [4,13,34]. Where *Y* stands for CPE constant, j is the imaginary number, *ω* is the angular frequency at which *Z* reaches its maximum value, and *n* is the exponent:(19)$$Z_{\mathrm{CPE}}=\frac{1}{Y(j\omega {)}^{n}}$$

While the values of the double-layer capacity were calculated by the following equations [3,4,27]:(20)$$C_{\mathrm{dl}}=\sqrt[{n}]{{YR}_{\mathrm{ct}}^{(1-n)}}$$

Table 5 shows the fitting parameters, and, from the values of χ^2^, it follows that the EEC used was the most appropriate one. The value of *R*_ct_ gradually increases with an increase in GG concentration, reflecting a maximum inhibition efficiency of 76.18 (i.e., at 0.3 g L^−1^) at 25 °C. In agreement with the results presented, the inhibition efficiency decreases, although slightly, with the temperature (i.e., 73.78% at 0.3 g L^−1^). It can also be seen that *C*_dl_ has the opposite trend of *R*_ct_. As reported by server studies, a decrease in *C*_dl_ values may be due to the decrease of the local dielectric constant and/or to an increase in thickness of the electrical double-layer according to the Helmholtz model [4,28,31,35]:(21)$$C_{\mathrm{dl}}=\frac{\varepsilon \varepsilon_{o}}{d}A$$
where *ε*_o_ is the permittivity of free space, ε is the local dielectric constant of the medium, *d* is the thickness of the protective layer, and *A* is the surface area of the electrode. The change of the local dielectric constant is generally assumed to be ascribed to the replacement of water molecules (which have a higher dielectric constant) by the inhibitor molecules (which have a lower dielectric constant) that form a protective layer onto the metal surface [6,26,27,28]. The presence of a second peak in the phase angle plot and the change of parameter *n* after the addition of GG seems to corroborate this assumption. It follows from the data that *R*_f_ increases with GG amount, which indicates that GG adsorbs on the N80 carbon steel surface and hinders the sweet corrosion of the metal via an inhibitor layer that varies the dielectric properties of the metal–brine solution interface [3]. The parameter *n* is a measure of the smoothness of the metal surface. The lower value of *n*_dl_ in the uninhibited solution signifies surface inhomogeneity due to the metal surface roughening and/or the presence of corrosion products [4,28]. On the other hand, after the addition of GG, the *n*_dl_ values steadily grow with the inhibitor concentration, which is an indication of a smoother surface due to its adsorption on the adsorption sites onto the steel surface, as shown in the surface analysis reported in Section 2.3.

The effect of the temperature can be summarized as follows: the *R*_ct_ increases as the temperature increases, but it decreases compared to the lower temperature, with a maximum inhibition efficiency of 75.87% (i.e., at 0.4 g L^−1^). The solubility of GG increases with the temperature, but its protectiveness decreases with the temperature. This may be ascribed to the increasing diffusion rate of corrosive spices that helps the transfer of electrons and, hence, increases the metal dissolution [5]. Also, *R*_f_ values are strongly reduced at higher temperatures, which can be explained by a shift of the adsorption–desorption inhibitor equilibrium process towards the desorption of GG from the metal surface, resulting in a decrease in its surface and protection [12,26,27,31].

### 2.2. Effect of Immersion Time

Figure 6 and Figure 7 show the EIS plots after different immersion times in the tested solution without and with 0.4 g L^−1^ of GG, respectively, at 25 and 50 °C. The plots have been fitted with EECs presented in Figure 5, and the calculated parameters are listed in Table 6. The figures show that, in the presence of GG, the size of the semicircles did not change significantly with different immersion times, which indicates that the inhibitor is quite stable at both temperatures. Also, it can be seen from the table that GG already shows its effectiveness in the early hours of the experiment (i.e., 6 h) at both temperatures. These results indicate fast adsorption of GG, which ensures immediate protection of the metal surface. At higher temperatures, the corrosion rate of the metal increases as shown by the decrease in the *R*_ct_ values for the sample exposed in the uninhibited solution. By contrast, upon the addition of GG, the IE% increases steadily, reaching a maximum value of 75.87%. It is worth mentioning that *R*_ct_ values in the presence of GG, in both temperatures, are stable for the whole duration of the experiment. This behavior may infer that GG immediately blocked the active sites on the surface in the first six hours of the experiment, and the corrosion process became stable.

### 2.3. Surface Analysis

FTIR spectroscopy was used to gain important information regarding the mode of interaction between the functional groups of the GG macromolecules and the metal surface. The stack plotted FTIR spectra of GG and the adsorbed GG on the metal surface after 24 h of immersion are shown in Figure 8a. The spectrum of the pure GG displays a peak at 3359 cm^−1^ which is attributed to the extensive intramolecular hydrogen bonds and stretching vibration of the -OH groups, and the peak at 2930 cm^−1^ is ascribed to the antisymmetric and symmetric C-H stretching vibrations of the methylene hydroxyl groups (-CH_2_OH) [7,13,15,37,38]. The spectrum of the steel surface adsorbed GG sample shows narrowed and shifted peaks at the same frequencies. The literature reported that this peaks-changing phenomenon may be due to the binding of some of the hydroxyl groups of the GG inhibitor with the metal surface via H-bond formation [7,13,15]. The region between 1800 and 800 cm^−1^ is considered to be the fingerprint region of polysaccharides. In brief, the peaks in the region between 1500 and 1200 cm^−1^ are correlated with twisting, bending/scissoring, and rocking vibrations of the C-H, whereas the region between 1200 and 800 cm^−1^ are correlated with the glucoside (1–4) linkage of galactose and (1–6) linkage of mannose and can be ascribed to the asymmetrical and symmetrical ring breathing vibrations (i.e., C-C-O, C-OH, and C-O-C) (Figure 8b) [7,13,15,37,38,39]. As observed from Figure 8a, the spectrum of GG adsorbed on the metal surface displays similar features, but the peaks are either shifted and/or narrowed around the same frequency range, which is evidence of possible adsorption of the tested inhibitor on the metal surface. The data are in agreement with the literature concerning the adsorption of gum compounds on the steel surface [4,7,13,15,38].

SEM-EDS and AFM analysis of samples immersed for 24 h in the tested solution without and with the inhibitor at 25 and 50 °C were carried out to complete the results observed by the gravimetric and electrochemical measurements. The SEM analysis in Figure 9a shows that, in the absence of GG, the surface of the steel appears damaged, with the cementite phase protruding from the surface after the preferential dissolution of the ferritic phase [4,35]. With time, Fe_3_C accumulates and then forms a layer of cementite on the metal surface, as observed from the EDS analysis shown in Figure 9b. The results show that the metal surface is covered by a layer mainly comprised of carbon and iron, plus other alloying elements from the sample microstructure, and only 3.7 wt.% of oxygen from the oxide corrosion product. By contrast, the surface of the sample immersed in the inhibited medium appears much smoother with some abrasive scratches (Figure 9c). The EDS analysis (Figure 9d) shows that, in the presence of GG, the concentration of carbon and oxygen elements is much higher compared to the uninhibited one. Notably, carbon and oxygen are part of the chemical composition of GG, hence it can be inferred that GG, after its adsorption, can form a protective layer that is capable of mitigating sweet corrosion, in agreement with the findings reported in the literature [4,15,40]. As the temperature increases, the corrosion rate also increases, and both surfaces appear damaged (Figure 10a,c) even though, in the presence of GG, the surface of the metal appears less damaged, with the presence of scratches resulting from the polishing process still visible. The EDS analysis (Figure 10b,d) is comparable to that obtained at lower temperatures. The percentage of carbon and oxygen is higher in the presence of GG, but much lower compared to the inhibited sample at lower temperatures. These results seem to agree with those obtained from electrochemical measurements, in which the resistance of the protective layer decreases with the temperature. GG at higher temperatures is likely desorbed from the steel surface, leading to the formation of a less dense and compact protective layer. Nevertheless, the results show that GG is still able to mitigate CO_2_ at higher temperatures to some extent. No iron carbonate layer was found on the metal surface. These findings are very well in agreement with the previous results. The literature reports [41,42,43] that, in the presence of CO_2_ and at a temperature below 40 °C, the surface is mainly covered by a porous layer of cementite and some alloying elements of the steel with little traces of FeCO_3_. These previous works pointed out that, at this temperature, FeCO_3_ precipitates only at elevated pH and/or at a high concentration of Fe^2+^ ions.

Figure 11 shows the 3D images of the tested samples with the average roughness (*R*_a_) and the root-mean-square (*R*_q_) listed in Table 7. The figures clearly show differences in surface roughness for the metal samples without and with the inhibitor. The surface roughness of the steel surface decreases in the presence of GG after 24 h of immersion in both temperatures. The surfaces look smoother, which can be ascribed to the formation of a compact protective layer of inhibitor on the steel surface that hinders the corrosion of the metal. The results are in good agreement with those observed from the EIS analysis which show that the exponent *n*_dl_ increases in the presence of GG in solution.

### 2.4. Mechanism of Inhibition

The results from the weight loss, electrochemical, and surface analysis have revealed that GG is adsorbed on the metal surface, forming a protective layer that hinders the corrosion processes. The schematic model reported in Figure 12 can be used to describe the possible adsorption mechanisms that occur on the metal surface.

The findings have revealed that the adsorption process is mainly a physisorption process. The inhibitor molecules are likely to interact with the metal substrate via its protonated form with the help of the chlorine ions adsorbed on the metal surface (Equations (22)–(24)) as shown in Figure 12a [4,15]:(22)$$\mathrm{GG}+xH^{+}↔\;[\mathrm{GG}H_{x}{]}^{x+}$$
(23)$$Cl_{\left(\mathrm{sol}\right)}^{-}+H_{2}O_{\left(\mathrm{ads}\right)}↔\;Cl_{\left(\mathrm{ads}\right)}^{-}+H_{2}O_{\left(\mathrm{sol}\right)}$$
(24)$$\;Cl_{\left(\mathrm{ads}\right)}^{-}+[\mathrm{GG}H_{x}{]}^{x+}↔(Cl_{\left(\mathrm{ads}\right)}^{-}\;-\;[\mathrm{GG}H_{x}{]}^{x+}{)}_{\left(\mathrm{ads}\right)}$$

PDP measurements also indicate that the cathodic current density decreases after the addition of GG in the tested solution. As observed from the FT-IR analysis, the specific broad peak ascribed to intramolecular hydrogen bonding and hydroxyl groups stretching vibration in the inhibitor structure (i.e., 3300 cm^−1^) is shifted and the intensity is reduced after its adsorption on the metal surface. As indicated by several studies, this change in shape could be ascribed to the interaction of the hydroxyl groups of GG with H^+^ absorbed on the cathodic sites of the steel surface via hydrogen bond formation [7,13,15]. As a result, the H_2_ evolution (Equation (12)) is suppressed, as shown in Figure 12b.

The GG macromolecules have many adsorbing sites, such as the oxygen atom in the hydroxyl groups. The lone pair of the electrons of these atoms can transfer their electron pairs to the empty d-orbitals of Fe atoms on the metal surface, with chemical or coordinate bond formation between the two parties (Figure 12c). Abdallah [14] also suggested that the adsorption of GG might occur via the coordination of the oxygen atoms from the glycosidic C(1)-O-C(4) linkage and the mannose units with Fe^2+^ ions formed on anodic sites (Figure 12d). The changes observed of the peaks in the regions 1200–1000 and 800–900 cm^−1^ for the adsorbed GG on the metal surface (Figure 8a), and from the literature, seem to confirm the formation of a carbohydrate-metal complex.

## 3. Materials and Methods

### 3.1. Materials

This study was carried out on N80 carbon steel with the following composition of (weight %): C 0.39%, Cr 0.04%, Cu 0.26%, Si 0.26%, Mn 1.80%, V 0.19%, Ni 0.04%, Al 0.03%, and the remainder Fe. The specimens were ground with several grades of silicon-carbide papers and polished up to 1 µm. Finally, the specimens were ultrasonically washed with distilled water and absolute alcohol and dried in the oven. All experiments were performed in 0.5 M of potassium chloride (KCl) in the presence of different concentrations of GG and temperatures. The aggressive solution was made from analytical grade material KCl (Sigma-Aldrich, Poznań, Poland) and pure deionized water. GG was also obtained from Sigma-Aldrich.

All the experiments were carried out thrice and the average result was calculated.

### 3.2. Weight Loss Measurements

The weight loss experiments were conducted in the absence and presence of different concentrations of GG and temperatures (i.e., 25 and 50 °C) after exposing the samples for 24 h in the tested solution. Before weighing, the corrosion products were removed and the samples were ultrasonically washed with distilled water and absolute alcohol, and finally, were dried in the oven. The corrosion rate (CR) and inhibition efficiency percentage (IE%) were calculated using the following Equations (25) [18,33] and (26) [4,15]:(25)$$\mathrm{CR}\;(\mathrm{mm}\;y^{-1})=\frac{87.6\;∆m}{dAt}$$
(26)$$\mathrm{IE}\;(\%)=\frac{\mathrm{CR}\;-\;CR^{\mathrm{inh}}}{\mathrm{CR}}\;\times \;100$$
where Δ*m* is the weight loss, *d* is the density of the metal, *A* is the surface of the sample, and *t* is the immersion time. CR and CR^inh^ are the corrosion rates of the steel without and with GG, respectively.

### 3.3. Electrochemical Measurements

The electrochemical experiments were performed in a three-electrode system with the N80 carbon steel sample used as a working electrode (exposed surface was 0.50 cm^2^), a saturated calomel electrode as a reference electrode, and a platinum foil as a counter electrode (CE). The electrochemical measurements were carried out in a Gamry reference 600 potentiostat (Gamry Instruments, Warminster, PA, USA). All the electrochemical measurements were carried out without and with the addition of different concentrations of GG and temperatures (i.e., 25 and 50 °C). The impedance spectroscopy (EIS) was carried out over the frequency range of 100 kHz to 10 mHz and amplitude of 10 mV at open circuit potential (OCP) after prefixed immersion times (i.e., 6, 12, 18, and 24 h). The potentiodynamic polarization (PDP) measurements were performed in the potential range from ±0.4 V vs. OCP with a scan rate of 1 mV s^−1^ after holding the specimen for 24 h. The EIS data and the PDP parameters were determined by using Echem Analyst 5.21 software. The IE% was calculated from the polarization resistances (*R*_p_) and corrosion current density (*i*_corr_) values using the Equations (27) [4,10] and (28) [4,10]:(27)$$\mathrm{IE}\;(\%)=\frac{R_{p}^{\mathrm{inh}}-R_{p}}{R_{p}^{\mathrm{inh}}}\;\times \;100$$
where $R_{p}^{\mathrm{inh}}$ and *R*_p_ are the polarization resistance values in the presence and absence of the inhibitor, respectively.
(28)$$\mathrm{IE}\;(\%)=\frac{i_{\mathrm{corr}}-i_{\mathrm{corr}}^{\mathrm{inh}}}{i_{\mathrm{corr}}}\;\times \;100$$
where *i*_corr_ and $i_{\mathrm{corr}}^{\mathrm{inh}}$ are corrosion current values without and with the inhibitor, respectively.

### 3.4. Surface Analysis

The surface morphology of the samples was investigated by immersing them in the aggressive solution for 24 h without/with 0.4 g L^−1^ GG.

FTIR analysis was carried out using a Thermo Scientific (Indianapolis, IN, USA) Nicolet iS50 spectrophotometer equipped with an attenuated total reflectance (ATR) accessory (Indianapolis, IN, USA). The FTIR spectra were obtained at a spectral resolution of 4 cm^−1^ with 256 co-added scans over the range from 3600 cm^−1^ to 800 cm^−1^. SEM-EDS analysis was carried out by using a JEOL scanning electron microscope at 2000× magnification. Atomic force microscopy (AFM) topography images were obtained using a NanoIR2 Anasys device (Santa Barbara, CA, USA). AFM images were performed in tapping mode using a 75 kHz resonance frequency (cantilever with a spring constant of 3 N m^−1^), scan rate 0.5 Hz, and resolution of 300 pts in the x and y directions. The average roughness (*R*_a_) and the root-means-square (*R*_q_) were determined on the surface area of 10 × 10 μm^2^.

## 4. Conclusions

Weight loss, electrochemical, and surface analysis methods have been used to estimate the potential of guar gum as a possible eco-friendly inhibitor for N80 carbon steel corrosion in a CO_2_-saturated saline environment. The obtained results can be summarized in the following points:The thickening agent guar gum was found to be an efficient corrosion inhibitor for N80 carbon steel in the tested solution, and its anticorrosive properties increased with its concentration but decreased with the temperature.EIS measurements performed at different immersion times have shown that the *R*_ct_ did not change significantly in either temperature, which indicates that GG can protect the metal surface even after prolonged immersion times.Potentiodynamic polarization measurements have revealed that GG acted as a mixed-type inhibitor.The results obtained also proposed that the adsorption of guar gum followed Temkin and Dubinin–Radushkevich adsorption isotherms. The adsorption parameters indicated that GG was spontaneously adsorbed onto the steel surface, and that the adsorption occurred mainly via an electrostatic or physical process. The values of *E*_a_ and *Q*_ads_ further support this hypothesis.The FTIR measurements reveal that GG was strongly adsorbed on the metal surface.SEM-EDS confirmed the adsorption of the guar gum and the formation of an absorptive layer onto the metal surface. The AFM studies confirmed a noteworthy decrease in the roughness of the metal surface after the addition of GG to the corrosive solution.

## Figures and Tables

**Figure 1 ijms-24-12269-f001:**
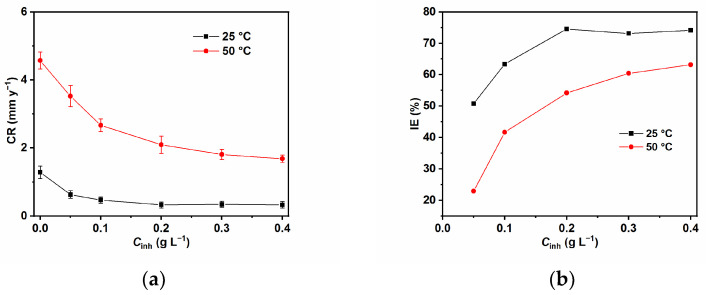
Corrosion rate (**a**) and inhibition efficiency (**b**) obtained from weight loss measurements at various concentrations of GG after 24 h of immersion at 25 and 50 °C.

**Figure 2 ijms-24-12269-f002:**
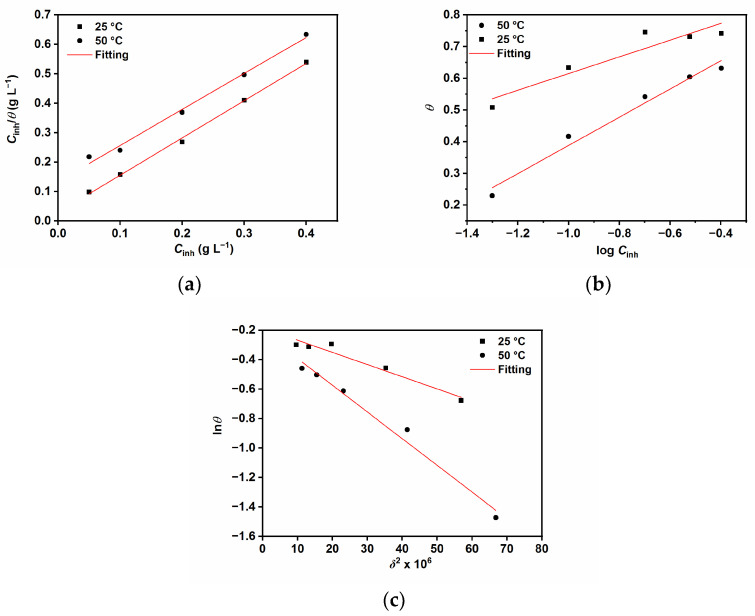
Adsorption isotherms (**a**) Langmuir, (**b**) Temkin, and (**c**) Dubinin–Radushkevich.

**Figure 3 ijms-24-12269-f003:**
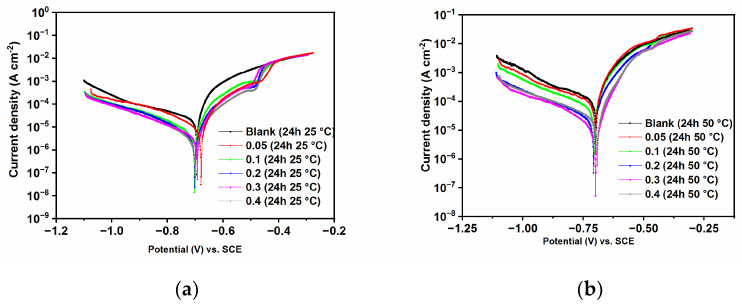
Potentiodynamic polarization measurements recorded without and with different concentrations of GG and different temperatures after 24 h of immersion time: (**a**) 25 °C and (**b**) 50 °C.

**Figure 4 ijms-24-12269-f004:**
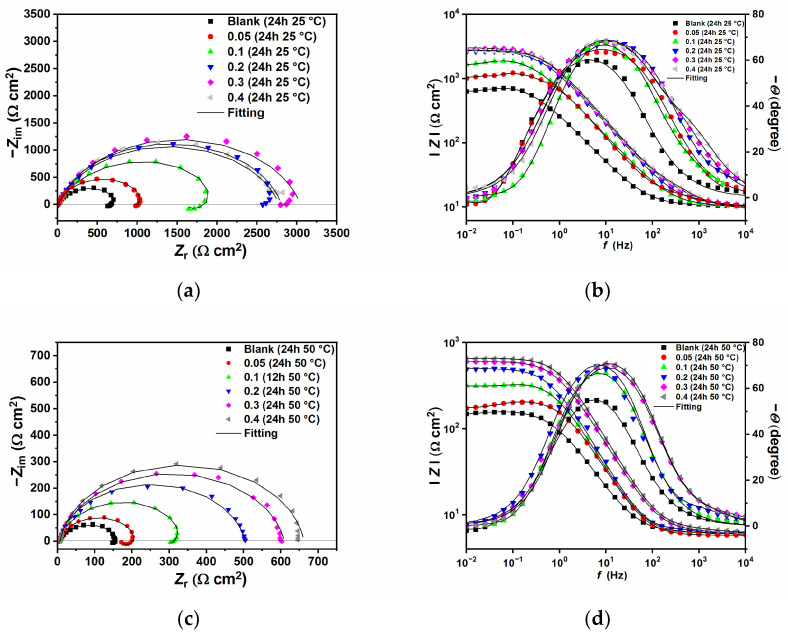
Nyquist and Bode plots obtained in the absence and presence of different concentrations of GG at 25 °C (**a**,**b**), and 50 °C (**c**,**d**).

**Figure 5 ijms-24-12269-f005:**
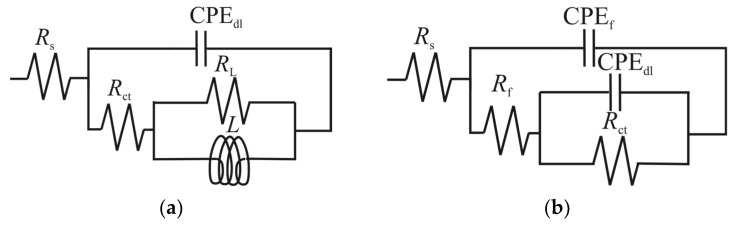
Equivalent circuits used to fit experimental data. (**a**) blank and lower concentrations of GG (e.g., up to 0.1 g L^−1^), (**b**) at higher concentrations of GG.

**Figure 6 ijms-24-12269-f006:**
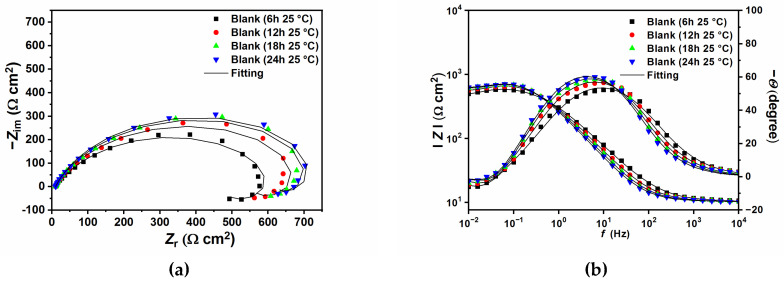
Nyquist and Bode plots recorded at open circuit potentials without (**a**,**b**) and with the presence of 0.4 g L^−1^ of GG (**c**,**d**) at 25 °C at different immersion times.

**Figure 7 ijms-24-12269-f007:**
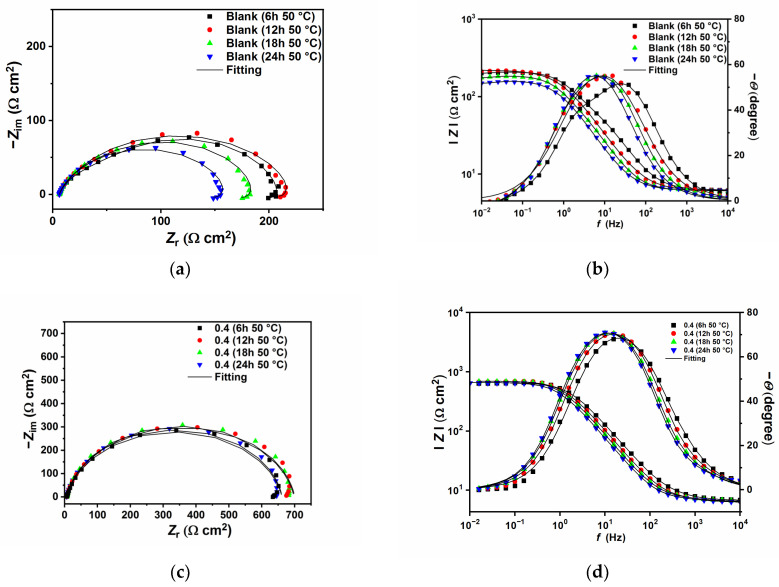
Nyquist and Bode plots recorded at open circuit potentials without (**a**,**b**) and with the presence of 0.4 g L^−1^ of GG (**c**,**d**) at 50 °C at different immersion times.

**Figure 8 ijms-24-12269-f008:**
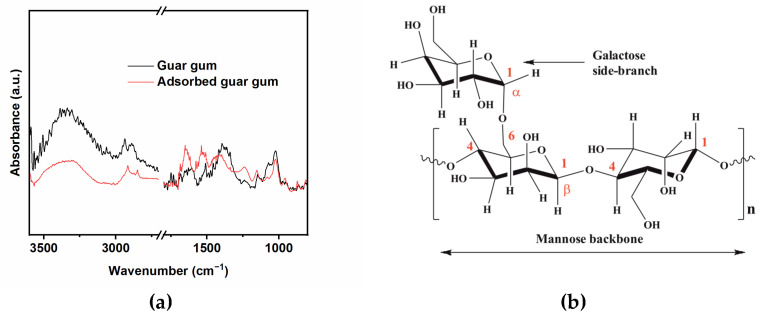
(**a**) FT-IR spectra of guar gum and surface-adsorbed guar gum. (**b**) chemical structure of guar gum.

**Figure 9 ijms-24-12269-f009:**
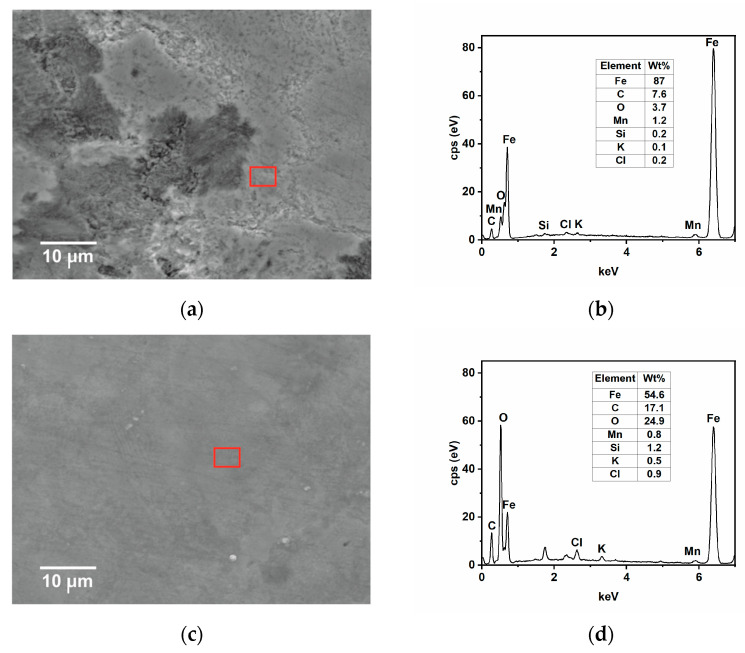
SEM and EDS (red square) analysis were recorded after 24 h of immersion in the tested solution, (**a**,**b**) without and (**c**,**d**) with the inhibitor (i.e., 0.4 g L^−1^) at 25 °C.

**Figure 10 ijms-24-12269-f010:**
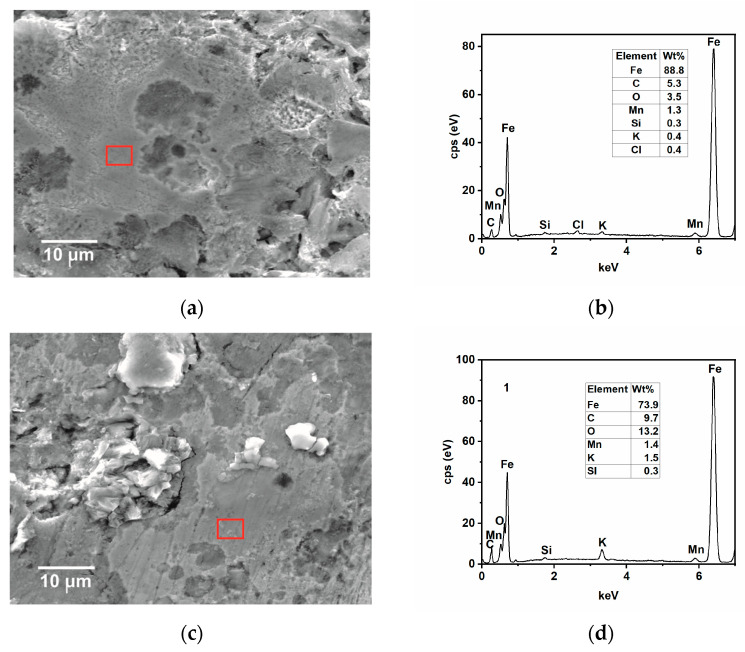
SEM and EDS (red square) analysis were recorded after 24 h of immersion in the tested solution, (**a**,**b**) without and (**c**,**d**) with the inhibitor (i.e., 0.4 g L^−1^) at 50 °C.

**Figure 11 ijms-24-12269-f011:**
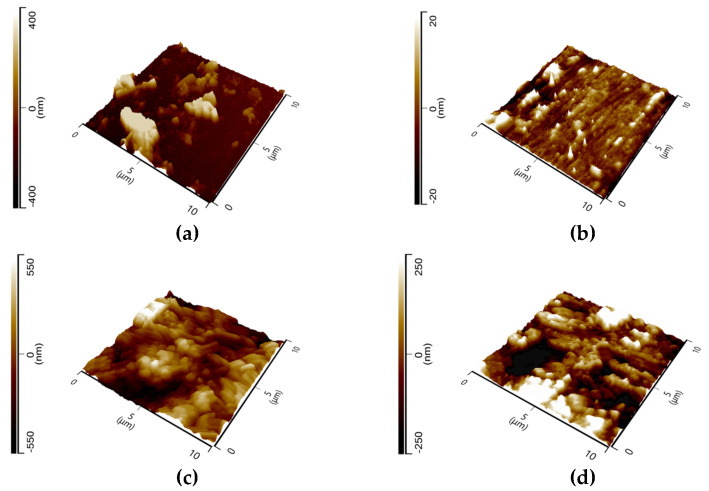
AFM micrographs of N80 carbon steel (**a**,**c**) immersed in the blank solution at 25 and 50 °C, respectively; (**b**,**d**) in the presence of 0.4 g L^−1^ of GG at 25 and 50 °C, respectively.

**Figure 12 ijms-24-12269-f012:**
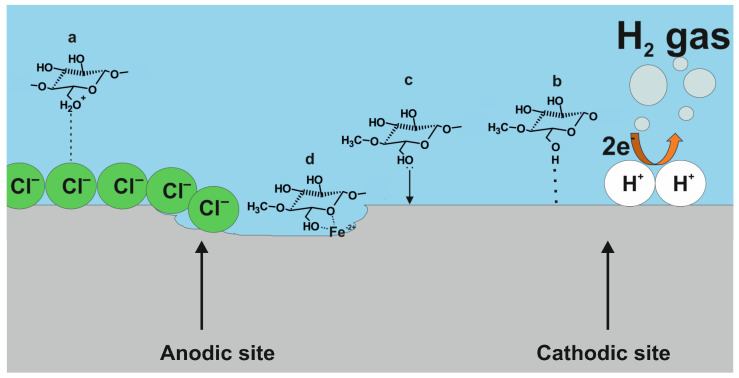
The possible mechanism of GG adsorption on N80 carbon steel. (**a**) electrostatic interaction, (**b**) H-bond, (**c**) chemical, and (**d**) chelating action.

**Table 1 ijms-24-12269-t001:** Comparison of reported inhibition efficiency of some other corrosion inhibitors originated from natural products with present inhibitor.

Inhibitors Derived from Natural Products	Metal Substrate	Corrosive Media	Inhibitor Conc. at Which Maximum Inhibition Efficiency Is Observed	Temperature (°C)	IE (%)	Time of Exposure (h)	Reference
Guar Gum	Carbon Steel	1 M H_2_SO_4_ containing NaCl	1.5 g L^−1^	25	93.88	24	[14]
Guar Gum	Carbon Steel	2.0 M H_3_PO_4_	1 g L^−1^	25	92.8	6	[13]
Xanthan gum	A1020 Carbon Steel	1 M HCl	1 g L^−1^	30	74.24	6	[24]
Gum Arabic	API 5L X42 steel	1 M HCl	2 g L^−1^	25	92	1	[25]
Gum Arabic	N80 Carbon Steel	0.5 M KCl (saturated in CO_2_)	0.5 g L^−1^	25	70.27	24	[4]
Chitosan	API 5L X60	3.5% NaCl (saturated in CO_2_)	0.1 g L^−1^	25	45.00	1	[11]
carboxymethyl cellulose	API 5L X60	3.5% NaCl (saturated in CO_2_)	0.1 g L^−1^	25	39.00	1	[11]

**Table 2 ijms-24-12269-t002:** Thermodynamic adsorption parameters of N80 carbon steel at 25 °C and 50 °C.

Temperature (°C)	*R* ^2^	*a*	*K*_ads_ (g L^−1^)	${∆G}_{\mathrm{ads}}^{{}^{\circ}}$ (kJ mol^−1^)	*ε* (mol^2^ kJ^−1^)	*E* (kJ mol^−1^)
Langmuir						
25	0.998	-	34.48	−25.90	-	-
50	0.992	-	7.41	−23.94	-	-
Temkin						
25	0.885	−4.38	2179.73	−36.18	-	-
50	0.977	−2.59	74.45	−30.14	-	-
Dubinin–Radushkevich						
25	0.951	-	-	-	0.008	7.76
50	0.982	-	-	-	0.018	5.27

**Table 3 ijms-24-12269-t003:** Apparent activation energies (*E*_a_) and heat of adsorption (*Q*_ads_) calculated in the temperature range of 25–50 °C.

*C*_inh_ (g L^−1^)	*E*_a_ (kJ mol^−1^)	*Q*_ads_ (kJ mol^−1^)
Blank	40.67	-
0.05	55.06	−39.90
0.1	55.55	−28.30
0.2	59.51	−29.06
0.3	53.10	−18.56
0.4	52.04	−16.52

**Table 4 ijms-24-12269-t004:** Potentiodynamic polarization parameters recorded without and with different concentrations of GG at 25 °C and 50 °C after 24 h of immersion time.

*C*_inh_ (g L^−1^)	*β*_c_ (mV dec^−1^)	*i*_corr_ (µA cm^−2^)	*−E*_corr_ (mV SCE^−1^)	IE (%)
25 °C				
Bank	390 ± 21	27.42 ± 0.03	694 ± 15	-
0.05	277 ± 17	15.50 ± 0.01	679 ± 11	43.47
0.1	221 ± 22	7.02 ± 0.02	709 ± 11	74.40
0.2	190 ± 11	4.69 ± 0.01	701 ± 16	82.90
0.3	207 ± 11	3.70 ± 0.01	693 ± 12	86.51
0.4	215 ± 9	4.20 ± 0.02	700 ± 10	84.68
50 °C				
Bank	414 ± 25	129.6 ± 0.12	696 ± 12	-
0.05	387 ± 12	99.96 ± 0.09	699 ± 16	22.86
0.1	318 ± 15	64.35 ± 0.03	704 ± 11	50.35
0.2	362 ± 15	32.10 ± 0.01	707 ± 11	75.23
0.3	349 ± 11	30.31 ± 0.02	699 ± 13	76.61
0.4	288 ± 12	24.57 ± 0.01	692 ± 12	81.04

**Table 5 ijms-24-12269-t005:** EIS parameters for the corrosion of N80 carbon steel in CO_2_-saturated saline solution carried out without and with different concentrations of GG at 25 and 50 °C.

*C*_inh_ (g L^−1^)	*R*_s_ (Ω cm^2^)	CPE_f_		*R*_f_ (Ω cm^2^)	CPE_dl_		*R*_ct_ (Ω cm^2^)	*L* (H cm^2^)	*R*_L_ (Ω cm^2^)	*C*_dl_ (mF cm^−2^)	*χ*^2^ (×10^−4^)	IE (%)
		*Y*_f_ (s^n^ mΩ^−1^ cm^−2^)	*n* _f_		*Y*_dl_ (s^n^ mΩ^−1^ cm^−2^)	*n* _dl_						
25 °C												
Bank	10.36 ± 0.95	-	-	-	0.770 ± 0.20	0.799 ± 0.12	617 ± 25	771 ± 105	105.40 ± 24.01	0.631	1.05	-
0.05	10.09 ± 0.98	-	-	-	0.440 ± 0.42	0.801 ± 0.11	929 ± 55	505 ± 50	95.51 ± 15.11	0.352	7.21	29.49
0.1	10.12 ± 0.91	-	-	-	0.286 ± 0.24	0.811 ± 0.12	1629 ± 89	2301 ± 200	580 ± 89.55	0.230	5.69	67.30
0.2	9.36 ± 0.56	0.04 ± 0.01	0.841 ± 0.12	13.43 ± 1.50	0.114 ± 0.13	0.833 ± 0.13	2755 ± 80	-	-	0.086	9.11	73.90
0.3	10.38 ± 0.88	0.09 ± 0.01	0.855 ± 0.11	46.38 ± 5.11	0.054 ± 0.01	0.843 ± 0.12	2986 ± 111	-	-	0.035	10.01	76.18
0.4	9.49 ± 0.78	0.064 ± 0.01	0.862 ± 0.15	34.62 ± 6.29	0.062 ± 0.02	0.862 ± 0.15	2761 ± 105	-	-	0.047	5.74	74.16
50 °C												
Bank	6.12 ± 0.25	-	-	-	1.645 ± 0.55	0.800 ± 0.05	139.90 ± 32.28	121.6 ± 10.12	18.35 ± 1.74	1.139	4.65	-
0.05	6.17 ± 0.36	-	-	-	0.776 ± 0.81	0.895 ± 0.61	168.20 ± 20.15	113.5 ± 11.00	41.94 ± 19.21	0.603	1.72	24.69
0.1	6.17 ± 0.59	-	-	-	0.697 ± 0.48	0.905 ± 0.19	302.50 ± 18.50	75.33 ± 35.17	39.69 ± 15.26	0.589	1.99	58.73
0.2	5.60± 0.48	0.39 ± 0.10	0.839 ± 0.25	4.49 ± 0.55	0.225 ± 0.13	0.900 ± 0.15	494.71 ± 30.51	-	-	0.176	5.39	68.30
0.3	5.65 ± 0.55	0.08 ± 0.01	0.910 ± 0.15	3.46 ± 0.31	0.292 ± 0.22	0.886 ± 0.27	599.00 ± 38.35	-	-	0.235	5.38	73.78
0.4	6.25 ± 0.57	0.12 ± 0.01	0.929 ± 0.11	4.49 ± 0.47	0.176 ± 0.19	0.906 ± 0.13	651.30 ± 20.88	-	-	0.142	2.96	75.87

**Table 6 ijms-24-12269-t006:** EIS parameters for the corrosion of N80 carbon steel in CO_2_-saturated saline solution were carried out at different exposure times 25 and 50 °C.

*C*_inh_ (g L^−1^)	Time (h)	*R*_s_ (Ω cm^2^)	CPE_f_		*R*_f_ (Ω cm^2^)	CPE_dl_		*R*_ct_ (Ω cm^2^)	*L* (H cm^2^)	*R*_L_ (Ω cm^2^)	*C*_dl_ (mF cm^−2^)	*χ*^2^ (×10^−4^)	IE (%)
			*Y*_f_ (s^n^ mΩ^−1^ cm^−2^)	*n* _f_		*Y*_dl_ (s^n^ mΩ^−1^ cm^−2^)	*n* _dl_						
25 °C													
Bank	6	10.31 ± 1.25	-	-	-	0.642 ± 0.25	0.733 ± 0.05	462.90 ± 23	118.40 ± 40	117.60 ± 44.31	0.412	1.19	-
	12	10.25 ± 1.05	-	-	-	0.692 ± 0.26	0.766 ± 0.02	549.20 ± 33	910.90 ± 88	107.60 ± 24.55	0.515	1.42	-
	18	10.24 ± 1.15	-	-	-	0.741 ± 0.29	0.784 ± 0.03	588.60 ± 29	786.30 ± 110	117.40 ± 21.21	0.589	1.52	-
	24	10.36 ± 0.95	-	-	-	0.770 ± 0.20	0.799 ± 0.02	617.00 ± 25	771 ± 105	105.40 ± 24.01	0.631	1.05	-
0.4	6	10.73 ± 1.18	0.125 ± 0.25	0.794 ± 0.19	33.65 ± 5.55	0.112± 0.52	0.827 ± 0.15	2582 ± 175	-	-	0.086	9.84	77.81
	12	9.84 ± 1.08	0.661 ± 0.02	0.855 ± 0.11	46.65 ± 5.13	0.044 ± 0.02	0.859 ± 0.25	2876 ± 125	-	-	0.031	3.37	77.53
	18	9.53 ± 0.95	0.633 ± 0.01	0.864 ± 0.01	41.78 ± 5.24	0.054 ± 0.02	0.864 ± 0.17	2795 ± 135	-	-	0.040	5.33	75.11
	24	9.49 ± 0.78	0.064 ± 0.01	0.862 ± 0.15	34.62 ± 6.29	0.062 ± 0.02	0.862 ± 0.15	2761 ± 105	-	-	0.047	5.74	74.16
50 °C													
Bank	6	5.96 ± 0.21	-	-	-	0.914 ± 0.20	0.76 ± 0.12	192.30 ± 12.28	123.10 ± 30.11	19.59 ± 1.21	0.526	9.55	-
	12	6.04 ± 0.15	-	-	-	1.168 ± 0.033	0.79 ± 0.11	200.90 ± 29.45	148.60 ± 26.25	18.38 ± 1.38	0.809	9.01	-
	18	6.00 ± 0.18	-	-	-	1.426 ± 0.67	0.82 ± 0.09	166.80 ± 23.22	126.60 ± 19.02	20.31 ± 1.55	0.996	5.55	-
	24	6.12 ± 0.25	-	-	-	1.645 ± 0.55	0.800 ± 0.05	139.90 ± 32.28	121.6 ± 10.12	18.35 ± 1.74	1.139	4.65	-
0.4	6	6.82 ± 0.75	0.131 ± 0.01	0.905 ± 0.16	6.31 ± 0.77	0.056 ± 0.05	0.879 ± 0.11	368.30 ± 32.41	-	-	0.035	9.62	43.44
	12	6.34 ± 0.30	0.128 ± 0.02	0.916 ± 0.10	4.43 ± 0.59	0.097 ± 0.09	0.895 ± 0.16	687.40 ± 29.26	-	-	0.041	4.50	68.33
	18	6.55 ± 0.48	0.139 ± 0.02	0.925 ± 0.09	4.18 ± 0.31	0.123 ± 0.11	0.900 ± 0.11	689.70 ± 25.77	-	-	0.093	3.00	73.03
	24	6.25 ± 0.57	0.123 ± 0.01	0.929 ± 0.11	4.49 ± 0.47	0.176 ± 0.19	0.906 ± 0.13	651.30 ± 20.88	-	-	0.141	2.96	75.87

**Table 7 ijms-24-12269-t007:** AFM parameters of the tested samples.

*C*_inh_ (g L^−1^)	*R*_a_ (nm)	*R*_q_ (nm)
25 °C		
Blank	97.41	149
0.4	5.62	7.95
50 °C		
Blank	156.14	198.55
0.4	116.88	154.06

## Data Availability

The data presented in this study are available on request from the corresponding author.

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
