# Peer review of "Guar Gum as an Eco-Friendly Corrosion Inhibitor for N80 Carbon Steel under Sweet Environment in Saline Solution: Electrochemical, Surface, and Spectroscopic Studies"

_ijms, 2023, doi:10.3390/ijms241512269_

Round 1

Reviewer 1 Report

Reviewer’s comments

Manuscript Number: ijms-2507494

Title: Guar Gum as an Eco-Friendly Corrosion Inhibitor for N80 Carbon Steel Under Sweet Environment in Saline Solution: Electrochemical, Surface and Spectroscopic Studies

Journal: International Journal of Molecular Sciences

The authors conducted a study to evaluate the corrosion inhibition of carbon steel in under sweet environment in a saline solution using guar gum. However, I have several comments as follows:

  1. All mathematical equations (1, 2, 3, 4, ….) should be supported with relevant references to enhance their credibility.

2.     It is necessary to provide characterization data of the guar gum inhibitor (e.g., FTIR, 13C NMR, 1H NMR, etc.).

3.     The abstract needs to be more informative, focusing on the novelty and key findings of the research. It should include the values of the main findings.

4.     The correlation between the structural/morphological findings and the electrochemical performance should be discussed and compared with other relevant materials. The following literature works could be referenced for this purpose: Journal of Chemical Sciences, 134 (2022), 90; Progress in Organic Coatings, 134 (2019), 272-280; Journal of the Iranian Chemical Society, 18(5) (2021), 1231-1241; Journal of Molecular Structure, 1242 (2021), 130831.

  1. The introduction should emphasize the novelty of the inhibitor used in this study.
  2. The inhibition efficiency of the guar gum inhibitor in the same media and on the same metal surface should be compared with previous reports.

7.     High-quality figures are required to enhance the clarity of the manuscript.

8.     It is recommended to study the reactivity of molecules using classic DFT, this will provide an in-depth understanding of the inhibition mechanism. 

Moderate editing of English language required

Author Response

Dear Editor,

Thank you very much for your kind interest in the editing and consideration of our manuscript. We are also very thankful to the reviewers of the manuscript for their high clarity in the reviews and for their valuable comments and suggestions toward the correction and improvement of the quality of the manuscript. According to the editor’s and reviewers’ comments, the revisions made were highlighted in the text in yellow. Also, the detailed corrections and answers to the questions are listed below point by point:

Response to the reviewer’s comments

The authors conducted a study to evaluate the corrosion inhibition of carbon steel in under sweet environment in a saline solution using guar gum. However, I have several comments as follows:

  1. All mathematical equations (1, 2, 3, 4, ….) should be supported with relevant references to enhance their credibility.

Answer: references have been added to the equations.

  1. It is necessary to provide characterization data of the guar gum inhibitor (e.g., FTIR, 13C NMR, 1H NMR, etc.).

Answer: The tested inhibitor is a commercial compound with a well-known structure, therefore, the other techniques suggested by the reviewer were not considered. We performed only the FTIR analysis on the native guar gum as reported in Figure 8a, to compare its spectrum with the one obtained from the adsorbed metal.

  1. The abstract needs to be more informative, focusing on the novelty and key findings of the research. It should include the values of the main findings.

Answer: the abstract has been improved.

  1. The correlation between the structural/morphological findings and the electrochemical performance should be discussed and compared with other relevant materials. The following literature works could be referenced for this purpose: Journal of Chemical Sciences, 134 (2022), 90; Progress in Organic Coatings, 134 (2019), 272-280; Journal of the Iranian Chemical Society, 18(5) (2021), 1231-1241; Journal of Molecular Structure, 1242 (2021), 130831.

Answer: a table with the other natural polysaccharides-like corrosion inhibitor has been added to the manuscript.

  1. The introduction should emphasize the novelty of the inhibitor used in this study.

Answer: the introduction has been modified to better describe the novelty of the study.

  1. The inhibition efficiency of the guar gum inhibitor in the same media and on the same metal surface should be compared with previous reports.

Answer: a table with the other natural polysaccharides-like corrosion inhibitor has been added to the manuscript

  1. High-quality figures are required to enhance the clarity of the manuscript.

Answer: I had to decrease the resolution (dpi) of the figures to reduce the size of the manuscript. Could you specify which one in particular needs to be improved, we can try to increase the resolution.

  1. It is recommended to study the reactivity of molecules using classic DFT, this will provide an in-depth understanding of the inhibition mechanism

Answer: we didn’t perform any density functional theory since the scope of this study was more practical. Unlike other natural compounds so far studied as corrosion inhibitors for sweet corrosion, this compound is already used in drilling fluids as a thickening agent, therefore the information that can be drawn can be used by scientists working in the oil and gas industry.

Reviewer 2 Report

This article focuses on the study of guar gum as a corrosion inhibitor for N80 carbon steel. Guar gum is a promising substance for a variety of processes. The authors proposed an interesting idea. There is an advantage of this work - the idea and the amount of data, as well as relevance. There are also some things that need to be improved:

1. Abstract needs to be expanded.

2. It is desirable to compare the experimental data with the literature data in more detail. This will give more validity to the conclusions and conclusions of the authors.

3. Please cite: 10.1007/s13399-021-01895-y. The FTIR part can be compared with the data from this work.

4. Why didn't the authors also choose the Elovich model for calculations?

5. It is desirable to put more emphasis on the obvious practical significance of the data obtained.

6. How firmly is guar gum attached to the surface?

7. Conclusions is desirable to expand.

Author Response

Dear Editor,

Thank you very much for your kind interest in the editing and consideration of our manuscript. We are also very thankful to the reviewers of the manuscript for their high clarity in the reviews and for their valuable comments and suggestions toward the correction and improvement of the quality of the manuscript. According to the editor’s and reviewers’ comments, the revisions made were highlighted in the text in yellow. Also, the detailed corrections and answers to the questions are listed below point by point:

Response to the reviewer’s comments

This article focuses on the study of guar gum as a corrosion inhibitor for N80 carbon steel. Guar gum is a promising substance for a variety of processes. The authors proposed an interesting idea. There is an advantage of this work - the idea and the amount of data, as well as relevance. There are also some things that need to be improved:

  1. Abstract needs to be expanded.

Answer: the abstract has been improved.

  1. It is desirable to compare the experimental data with the literature data in more detail. This will give more validity to the conclusions and conclusions of the authors.

Answer: a table with the other natural polysaccharides-like corrosion inhibitor has been added to the manuscript

  1. Please cite: 10.1007/s13399-021-01895-y. The FTIR part can be compared with the data from this work.

Answer: the reference has been added and compared with the data obtained.

  1. Why didn't the authors also choose the Elovich model for calculations?

Answer: the isotherm used in this manuscript to characterize the tested inhibitor was mainly selected from the equation used in literature to characterize other polysaccharides-like corrosion inhibitors.

  1. It is desirable to put more emphasis on the obvious practical significance of the data obtained.

Answer: the introduction has been modified to better describe the novelty of the study.

  1. How firmly is guar gum attached to the surface?

Answer: the adsorption of GG occurred mainly via an electrostatic or physical adsorption process; therefore, it can be said it is not strongly attached to the metal surface.

  1. Conclusions is desirable to expand.

Answer: the abstract has been improved.

Round 2

Reviewer 1 Report

Reviewer’s comments

Manuscript Number: ijms-2507494

Title: Guar Gum as an Eco-Friendly Corrosion Inhibitor for N80 Carbon Steel Under Sweet Environment in Saline Solution: Electrochemical, Surface and Spectroscopic Studies

Journal: International Journal of Molecular Sciences

The authors conducted a study to evaluate the corrosion inhibition of carbon steel in under sweet environment in saline solution using guar gum. However, I have several comments as follows:

  1. Although there are a lot of data, the novelty of this study is totally missing. Where there is no effort done for modification the inhibitor as well as the same inhibitor has been used for the same material (Carbon Steel), as in Refs. 13 and 14.

2.     Although the guar gum inhibitor is commercial compound, it is necessary to provide characterization data (e.g., FTIR, 13C NMR, 1H NMR, etc.). This is very important to understand and justify the inhibition behavior and mechanism.

3.     The correlation between the structural/morphological findings and the electrochemical performance should be discussed.

4.     It is recommended to study the reactivity of molecules using classic DFT, this will provide in-depth understanding of the inhibition mechanism. 

 Moderate editing of English language required

Author Response

Dear Editor,

Thank you very much for your kind interest in the editing and consideration of our manuscript. We are also very thankful to the reviewers of the manuscript for their high clarity in the reviews and for their valuable comments and suggestions toward the correction and improvement of the quality of the manuscript. According to the editor’s and reviewers’ comments, the revisions made were highlighted in the text in yellow. Also, the detailed corrections and answers to the questions are listed below point by point:

Response to the reviewer’s comments

The authors conducted a study to evaluate the corrosion inhibition of carbon steel in under sweet environment in a saline solution using guar gum. However, I have several comments as follows:

  1. Although there are a lot of data, the novelty of this study is totally missing. Where there is no effort done for modification the inhibitor as well as the same inhibitor has been used for the same material (Carbon Steel), as in Refs. 13 and 14.

Answer: The introduction has been modified to better clarify the novelty of this study. The referenced 13 and 14 used GG as a corrosion inhibitor but were performed in strong acid solution (e.g., 1M H2SO4, 2M H3PO3, 1M HCl, etc.). and for short immersion times. The corrosion of the steel in the presence of CO2 is more complex. The formation of iron carbonate on the metal surface can influence the corrosion resistance of the steel.

  1. Although the guar gum inhibitor is commercial compound, it is necessary to provide characterization data (e.g., FTIR, 13C NMR, 1H NMR, etc.). This is very important to understand and justify the inhibition behavior and mechanism.

Answer: As mentioned in the first round of revision, we think that these analyses are not necessary since the commercial compound has a well-known structure. Also, the publication mentioned by the reviewer (i.e., refs 13 and 14) or other publications referenced in this manuscript (e.g., 9, 11, 12, etc.) that used a commercial compound as a corrosion inhibitor didn’t report any above-mentioned analysis.

  1. The correlation between the structural/morphological findings and the electrochemical performance should be discussed.

Answer: FT-IR analysis has been modified so as also the interaction of the intramolecular hydrogen bonds and stretching vibration of the -OH groups and the antisymmetric and symmetric C-H stretching vibrations of the methylene hydroxyl groups (-CH2OH) are discussed.

  1. It is recommended to study the reactivity of molecules using classic DFT, this will provide in-depth understanding of the inhibition mechanism. 

Answer: As mentioned in the first revision, the scope of this manuscript is more practical. Compared to the other natural compounds so far studied as corrosion inhibitors for sweet corrosion, which most likely will never be practically used because the inhibition efficiency is too low compared to a commercial one, GG is already used in the drilling fluid. It is used as a thickening agent, but the information presented in this manuscript shows that can be also practically used as a corrosion inhibitor for sweet corrosion.

Round 3

Reviewer 1 Report

The authors conducted a study to evaluate the corrosion inhibition of carbon steel in under sweet environment in saline solution using guar gum. The authors have addressed most of the comments in the revised version. The current version could be accepted for publication.

Minor editing of English language required